# Pregnancy-Associated Breast Cancer: A Diagnostic and Therapeutic Challenge

**DOI:** 10.3390/diagnostics13040604

**Published:** 2023-02-07

**Authors:** Francesca Galati, Valentina Magri, Paula Andrea Arias-Cadena, Giuliana Moffa, Veronica Rizzo, Marcella Pasculli, Andrea Botticelli, Federica Pediconi

**Affiliations:** 1Department of Radiological, Oncological and Pathological Sciences, Sapienza-University of Rome, 00161 Rome, Italy; 2Clinica Imbanaco, Cali 760031, Colombia

**Keywords:** PABC, pregnancy, lactation, breast imaging, breast cancer treatment

## Abstract

Pregnancy-associated breast cancer (PABC) is commonly defined as a breast cancer occurring during pregnancy, throughout 1 year postpartum, or during lactation. Despite being a rare circumstance, PABC is one of the most common types of malignancies occurring during pregnancy and lactation, with growing incidence in developed countries, due both to decreasing age at onset of breast cancer and to increasing maternal age. Diagnosis and management of malignancy in the prenatal and postnatal settings are challenging for practitioners, as the structural and functional changes that the breast undergoes may be misleading for both the radiologist and the clinician. Furthermore, safety concerns for the mother and child, as well as psychological aspects in this unique and delicate condition, need to be constantly considered. In this comprehensive review, clinical, diagnostic, and therapeutic aspects of PABC (including surgery, chemotherapy and other systemic treatments, and radiotherapy) are presented and fully discussed, based on medical literature, current international clinical guidelines, and systematic practice.

## 1. Introduction

All breast disorders emerging during pregnancy and/or lactation should be “handled with care”. On the one hand, hormone-induced anatomical and functional changes occurring in breast tissue during lactogenesis may cause an overlap in imaging appearance of lesions, as well as in physical examination. On the other hand, pregnancy and postpartum are extremely delicate moments in a woman’s life, and psychological aspects of fear and anxiety, especially in a context of life changes, need to be always considered.

Breast cancer during pregnancy and lactation is rare, but delayed diagnosis is frequent, mainly due to lack of awareness of this clinical entity, fear of X-ray-based diagnostic examinations, as mammography, and, probably, a certain degree of denial of suspicious signs and symptoms. Nonetheless, a quick and precise clinical and radiological assessment is essential, as well as a fast multidisciplinary management.

Pregnancy-associated breast cancer (PABC) is often aggressive; therefore, the postponement in management until delivery or after the end of lactation should be avoided, as delay is associated with poor prognosis. 

In the present review, diagnostic and therapeutic aspects of PABC are presented and discussed, based on available literature, current international guidelines, and systematic clinical practice.

## 2. Background and Epidemiology

Breast cancer (BC), cervical cancer, hematological diseases, and melanoma are the most common malignancies developing during pregnancy and lactation, the former two accounting for 50% of gestational cancers, followed by hematological cancers that comprise a further 25% of gestational cancer cases [1]. Less common oncological diseases during pregnancy are ovarian, thyroid, and colon cancers [1,2]. PABC is commonly defined as a BC occurring during pregnancy, throughout 1 year postpartum, or during lactation. PABC affects about 1 in 3000 pregnant women [1,3], representing about 7% of all BCs in women under 45 years old, 10% in women under 40, and rising to 15.6% in women under 35 [4,5,6]. The reported incidence of PABC is 15–35 per 100,000 deliveries [7], and it is higher in the postpartum period, with the great majority of cases detected within six months postpartum [8].

Delayed diagnosis is frequent, mainly because physiological changes or small disorders related to pregnancy may mask initial signs or symptoms of BC [9]. 

The particular aggressiveness typical of this condition is explained by many reasons, such as the young age at diagnosis, the advanced T stage, and the high frequency of estrogen/progesterone receptors (ER/PR) negativity and/or Human Epidermal Growth Factor Receptor 2 (HER2) positivity [9,10]. Moreover, lymphovascular invasion and lymph nodes involvement are more common [1,3]. As a consequence, the clinical outcome is poorer and mortality is higher in women with PABC compared with nulliparous women [1,11]. In addition, the treatment of PABC can be limited or delayed to ensure fetal safety [9]. 

Most patients have no family history of BC, even if BRCA mutation carriers are at high risk of developing PABC [8]. 

## 3. Clinical Presentation

PABC mainly presents as a persistent breast mass. In a study including 142 patients [10], in 100% of patients, the first symptom was a self-discovered breast mass. This finding is consistent with most of the available literature [8,12,13]. Other symptoms include skin thickening and nipple discharge, either bloody or purulent [10,12,13]. The presence of lymph node involvement and inflammatory changes are frequent at diagnosis [8,10].

However, during pregnancy and lactation, there are several physiological changes of the breast tissue that may mask palpable masses and other BC features. In addition, the physical examination, either self-examination or medical examination, is particularly difficult [8]. Finally, the engorged and edematous appearance of breast tissues can be misleading in the detection of lesions, increasing both false negative and false positive rates (as purely physiologic findings could be interpreted as pathological) [8]. It should also be considered that young, premenopausal women do not undergo breast image screening frequently, making it more difficult to detect non-palpable breast masses [8]. According to the data found, we consider that the presence of palpable breast lumps, skin changes, or non-dairy nipple discharge in pregnant or lactating patients should not be a priori associated with their condition, but rather require a thorough clinical examination and imaging work-up, in order to reduce the delay in PABC diagnosis and treatment.

## 4. Imaging

Ultrasound (US) is the first-line imaging modality in women with suspected PABC, with reported high sensitivity and negative predictive values [8]. The typical appearance of PABC at US examination is a hypoechoic mass with fluid component, related to central necrosis or cystic degeneration due to the outgrowing vascular supply, characterized by irregular margins and posterior shadowing [8]. Figure 1 shows an example of invasive ductal carcinoma in a 36-year-old woman at the 6th week of pregnancy. The lesion has the typical characteristics of malignancy (hypoechoic pattern, irregular margins, and posterior shadowing).

However, once again, physiological changes associated with pregnancy and lactation may alter typical US hallmarks. Therefore, the new onset of a palpable complex cystic mass during pregnancy and lactation needs tissue sampling [14] and should not be ignored or “downgraded” to a galactocele or an abscess, unless signs and symptoms are strongly suggestive or obvious.

The value of breast US during pregnancy goes beyond the mere detection of BC, as it can also provide information about nodal disease and response to neoadjuvant chemotherapy, and can guide biopsy procedures of suspicious lesions. When the histopathological result of tissue sampling is not consistent with radiological findings, surgical excision should be considered [15].

The role of mammography is relatively diminished during pregnancy and lactation, due to the concerns related to radiation exposure for the fetus and to the increased mammographic density of the breast parenchyma. As in non-pregnant women, mammography can visualize calcifications, masses, and architectural distortion [15] (Figure 2 and Figure 3), although the increased mammographic density may reduce the sensitivity of this technique [16].

Figure 2 shows a standard mammography of a 42-year-old woman during the 7th week of pregnancy. Both mediolateral oblique and craniocaudal views show a high-density irregular mass with surrounding parenchymal distortion and some contextual calcifications between the upper quadrants of the left breast.

Figure 3 shows the standard mammography of the right breast of a 36-year-old breastfeeding woman in the 6th month postpartum. In a context of general increased density due to breastfeeding, a voluminous, high-density mass with irregular margins is visible between the inner quadrants. In the mediolateral oblique view, axillary lymphadenopathies are also evident.

For what concerns radiation exposure, mammography is nowadays considered safe in pregnancy, as the radiation dose from a bilateral two-view standard mammogram is <3 mGy per view (equivalent to about 7 weeks of background radiation); moreover, an appropriate abdominal shielding allows a further reduction in the fetal radiation dose [17]. 

According to the latest American College of Radiology (ACR) appropriateness criteria, mammography is usually appropriate and should be used as an adjunct to US [18]. 

Lactating patients should be advised to nurse or pump immediately before mammography, in order to decrease parenchymal density related to the presence of milk [17]. 

There is no current concern about the increase in radiation dose when performing digital breast tomosynthesis in pregnant women. Similarly, there are no current recommendations regarding the restriction in using contrast-enhanced mammography in pregnancy and lactation. Nonetheless, limitations of this technique include: radiation dose exposure, the use of iodinated contrast medium (which increases the risk of hypothyroidism in the fetus when used after the 12th week of gestation), and a pregnancy-associated increase in background parenchymal enhancement (BPE), which is known to reduce diagnostic accuracy of contrast-enhanced breast imaging. 

Despite its evident diagnostic superiority, contrast-enhanced breast magnetic resonance imaging (CE-MRI) is not currently considered a safe modality during pregnancy, due to concerns regarding fetal exposure to gadolinium-based contrast agents, which are able to cross the placental barrier, and to the prone position during the examination [19,20]. 

As a consequence, experiences of breast MRI during pregnancy are currently limited to examinations performed before elected abortion or using unenhanced protocols [17].

In recent years, novel unenhanced functional techniques such as diffusion-weighted imaging (DWI) have been widely investigated to improve breast MRI accuracy [21,22,23], and DWI is having an increasing role as a valuable diagnostic tool during pregnancy and lactation, in order to facilitate an earlier diagnosis of PABC [24]. 

A recent study investigating the role of DWI as a stand-alone modality for pregnant patients at high risk or with newly diagnosed PABC has suggested a potential additive diagnostic role of DWI for a noninvasive approach in breast evaluation during pregnancy. Moreover, the same study has demonstrated that breast MRI in the prone position is feasible and tolerable during pregnancy, although most of the patients enrolled were in the first or second trimesters. [25]. 

DWI is also gaining recognition for systemic staging of PABC, when the use of PET/CT is contraindicated. In particular, whole-body MRI using DWI with a background suppression (DWIBS) sequence can provide non-invasive information regarding local staging and distant metastasis [26,27].

In the postpartum setting, CE-MRI of the breast can be performed, as gadolinium-based contrast media are considered safe during lactation [17]. Nevertheless, imaging interpretation may be limited due to the growth of fibroglandular tissue and to the increased vascularity of lactating breast tissue [28], which make the use of CE-MRI controversial. Indeed, the limited sensitivity related to the increased pregnancy-associated BPE may mask suspicious lesions or result in false positive findings [17,18,29].

A recent study has compared tumor conspicuity on CE-MRI and on diffusion tensor imaging (DTI) parametric maps among lactating patients with PABC [30]. Because of the marked BPE affecting lactating breasts, tumor conspicuity was reduced by 60% than non-lactating controls on CE-MRI, while a 138% increase in tumor conspicuity was observed on DTI compared with CE-MRI, highlighting a clear advantage of unenhanced techniques in the postpartum setting. On the contrary, DWI has demonstrated a lower sensitivity in cases of sub-centimeter lesions, as well as in non-mass-enhancement lesions [31,32]. 

However, breastfeeding discontinuation for 1–2 weeks is usually sufficient to decrease BPE levels. Likewise, in the post-weaning, lactation-related BPE significantly drops, and CE-MRI utility returns to the optimum level [17].

In conclusion, there is wide agreement that US should be the first-line modality in the diagnostic work-up of PABC. Mammography is safe with the proper precautions but is currently considered ancillary to US. The role of MRI in PABC work-up is still controversial due to concerns regarding the prone position and the fetal exposure to gadolinium-based contrast media. However, the development of MRI unenhanced functional techniques has the potential to increase the use of breast MRI during pregnancy and lactation, in order to facilitate an earlier diagnosis of PABC.

## 5. Histology

As in non-pregnant women, ductal invasive carcinoma is the most common histological type of PABC [8], representing 78–88% of cases [33,34], usually with low ER and PR expression [12]. Invasive lobular carcinoma and other less common histologic types have been found to be uncommon in patients with PABC [12] (Figure 4).

Both hormonal changes and transient immunosuppression during pregnancy have been considered responsible for the development of PABC. In addition, involution and other changes in breast tissue are considered important risk factors, as they share characteristics with a proinflammatory microenvironment [1,35].

The results of a systematic review of 14 case–control studies conducted by Marikakis et al. [36] found that hormone-receptors-negative cases were more frequent in PABC than in non-PABC patients.

PABC is frequently more aggressive, with a high histologic grade and a more advanced stage, at diagnosis [13]. Moreover, a higher incidence of inflammatory BC compared to non-pregnant women was found [4,33,37].

Termination of lactation induces a mammary remodeling, regulated by fibroblasts, endothelial cells, and immune cells [28]. The activation of these cells can lead to the growth and development of transformed cells [38]. An in vitro study has demonstrated how mammary involution can favor the growth of existing tumor cells [38]. 

A recent review [39] of studies about the genomic profile of PABC has shown aberrant expression of several oncogenes, tumor suppressor genes, apoptosis and transcription regulators, and genes involved in DNA repair mechanisms, in cell proliferation, in the immune response, and in other significant biological processes. However, the molecular nature of PABC remains partially unexplained and (the authors have concluded) more studies are required to formulate conclusions and recommendations regarding diagnosis, prognosis, and possible treatment.

Two pathways are enriched in PABC: the G protein-coupled receptor and the serotonin receptor pathway [36,40]. The up-regulation of serotonin can induce tumorigenesis by cellular proliferation [41]; moreover, the serotonin receptor pathway is also involved in the regulation of the expression of cathepsin S, and is highly expressed in several cancer subtypes [42].

HER2 expression is unclear, as authors have found negative expression during pregnancy and lactation, with positive expression after delivery or cessation of lactation [8]. This could be secondary to stable high estrogen levels during pregnancy that lead to a down-regulation of the expression of ER in some cell lines [43]. On the contrary, other authors have described overexpression [44]. Among these, Bae et al. [45] have evaluated 2810 cases of women under 40 years with BC, including 40 PABC and 2770 BC not associated with pregnancy, and have found that PABC had higher HER2 overexpression (38.5%). The possible explanation given by the authors is that this receptor plays an important role in embryogenesis and the development of some tissues such as lung, skin, gut, muscle, heart, and neural tissue [45,46]. Two more studies have found no differences in terms of HER2 overexpression between PABC patients and the control group [9,47]. Nevertheless, in one study, there was no difference in terms of BC-related family history between the two groups [47], while in the other study [9], only 8 out of 41 non-PABC were nulliparous, and 33 were diagnosed more than 1 year after delivery. The differences among PABC patients and the control group of each study could probably explain this finding.

Wilms’ tumor 1 (*WT1*) is a tumor suppressor gene identified in Wilms tumors [48] that participates in the embryogenesis of many organs by regulating multiple target genes and signaling pathways. Moreover, it was detected in some tumors, suggesting that WT1 could also function as an oncogene [9]. In BC patients, the expression of WT1 is associated with high histological grade, ER negativity, and HER2 subtype [9]. In PABC, WT1 expression was found substantially increased in vascular structures of the invasive cancer component, in comparison with that of non-PABC [44].

P63 is a member of the p53 family. It is expressed on ductal myoepithelial cells and lobules and is necessary for the development of the mammary gland. A study has shown that epithelial cells with aberrant WT-1 and p63 expression did not express ER or PR and had a higher proliferation index [49].

Ki-67 is a nuclear protein and a biomarker of cellular proliferation. There are several potential intended uses for Ki-67, and clinical utility must be determined for each [50]. Some studies have evaluated Ki-67 in PABC, showing that PABC exhibits a higher expression of Ki-67 in comparison to age-matched non-pregnant women [10,51]. 

In conclusion, most of the studies have found that PABC was more frequently high-graded, showed a low ER/PR expression, an HER2 overexpression, a higher expression of p53 and WT1, and had a higher Ki-67 proliferation index. 

New tools for the earlier detection of molecular biomarkers of carcinogenesis as well as for a better evaluation of cancer prognosis are expected from translational research in PABC.

MicroRNAs (miRNAs) are small non-coding RNAs that have been shown to regulate the signaling pathways and tumor biology in PABC. miRNA profiling of 56 tumors from women who developed cancer after delivery has demonstrated different expressions and different methylation patterns between tumors of the early versus late postpartum, suggesting the possibility of different epigenetic modulations of PABC genes in different phases of pregnancy and puerperium [52].

In another experience, the expression of miR-21 was evaluated in 25 patients with PABC. The overexpression of miR-21 was significantly associated with non-luminal cancer, with higher-grade tumors and positive axillary nodes in PABC patients versus non-PABC patients with similar features. Moreover, the expression of target proteins PTEN, BCL2, and PDCD4 was decreased in 80%, 76%, and 40% in PABC patients, respectively. These results indicate that overexpression of miR-21 may predict worse prognosis in PABC patients [53].

These studies, although preliminary and based on small sample sizes, have demonstrated that specific miRNAs could represent, in the future, prognostic and predictive biomarkers able to monitor the biologic evolution of cancer in PABC patients.

Recent technological and molecular advances have shown how new procedures, such as the so-called “liquid biopsy”, are able to provide the opportunity of very early detection, prognostication, and real-time drug-response monitoring through a minimally invasive procedure such as a blood test. 

In addition, the detection and characterization of circulating tumor cells (CTCs), circulating tumor DNA (ctDNA), and exosomes have opened enormous possibilities to assess tumor heterogeneity and the biological evolution of cancer. However, the use of these techniques is still limited and restricted to clinical research, due to the need of a better standardization of pre-analytical and analytical aspects, preventing, de facto, the investigation of their possible role in daily clinical practice [54].

## 6. Local and Systemic Treatments

### 6.1. Surgery

Surgery is considered the safest treatment at any stage of pregnancy. Several studies have shown that most anesthetic agents are safe for the fetus [55]. Furthermore, from the 24th-26th week of gestation onward, fetal conditions can be closely followed through intraoperative fetal heart rate monitoring. Cardiotocography is useful in the postoperative phase as analgesia can alter the identification of mild early contractions and delay tocolysis [56]. In addition, postoperative venous stasis and pregnancy-associated hypercoagulability increase the risk of deep venous thrombosis and pulmonary embolism, making heparin prophylaxis seriously needed. 

As the diagnosis of PABC is usually delayed, with a consequent greater tumor size at diagnosis, a breast preservation approach is often less feasible, making mastectomy preferred. Breast reconstruction is possible; however, due to physiological changes during pregnancy, an active debate is present in the literature about the timing of reconstruction: whether to delay reconstruction until after delivery or to implant an expander during pregnancy. 

The latter option has many advantages, including improved psychological and aesthetic results and a short operative time, without morbidity for the patient or the fetus [57,58]. In a paper by Lohsiriwat et al. [59], the non-inferiority of this approach was demonstrated in terms of obstetrical or surgical complications.

For the identification of sentinel nodes, radionuclide injection is safe and accurate in pregnant patients [60], whereas the teratogenic effects of Methylene Blue preclude its use, while little is known about the safety of Isosulfan Blue during pregnancy [61].

### 6.2. Radiotherapy

An international consensus meeting in 2010, involving experts from eight different European countries, concluded that radiotherapy can be relatively safe in the first and second trimesters of pregnancy [62]. However, the risks of fetal exposure to high doses of radiation are significant because the radiation dose during radiotherapy is greater than that received during diagnostic procedures [63].

The risk of fetal damage is mainly related to two variables: the radiation dose and the gestational age. For what concerns the dose, it has been estimated that malformations may occur if the in utero dose exceeds 0.2 Gy; therefore, 0.1–0.2 Gy is considered to be the safety threshold.

Regarding gestational age, excessive exposure during the first two weeks after conception may result in implantation failure or death (all-or-nothing law). During the 2nd-8th week after conception, malformations may occur, especially during the organogenesis stage. From the 8th to the 25th weeks, the central nervous system is extremely sensitive to radiation; thus, excessive exposure during this period may result in mental disability [64]. The long-term effects of radiation have not yet been fully enlightened. Therefore, this procedure is generally not recommended in routine medical practice during pregnancy [65]. The risk of developing cancer has been shown to be stochastic and proportional to radiation dose—appearing at 0.01 Gy—while the severity of the injury is unrelated. In addition, although supra-diaphragmatic irradiation results in a reduced dose to the uterus, the diaphragm becomes more elevated with increasing gestational age, increasing the risk of exposure.

Therefore, this therapeutic approach is usually recommended after delivery whenever possible. As starting adjuvant radiotherapy within 8–12 weeks after surgery has been shown to improve disease-free survival (DFS) and to lead to a lower risk of local recurrence in patients with BC, some studies have explored the possibility of considering it also during pregnancy, trying to minimize fetal exposure in carefully selected patients.

Deferring radiotherapy is a difficult decision to make, especially during the first trimester of pregnancy, when chemotherapy is also not recommended. Therefore, each case should be discussed by a multidisciplinary team including gynecologists and obstetricians, with careful consideration of the patient’s prognosis and gestational age. In any case, after a very careful counseling, the patient must make her own decision and the clinician should respect her right to self-determination [66].

### 6.3. Chemotherapy

The diagnosis of PABC has ethical and therapeutic implications for both patients and oncologists. Pregnant women with cancer have an innate desire to protect the fetus, while physicians must balance embryonic and fetal well-being with maternal prognosis. Most therapies can be safely administered to pregnant as well as non-pregnant patients. Unfortunately, knowledge in this field is limited, and many issues, such as timing of treatment, appropriate drug choice, total daily dose, and fraction, are still not clearly defined [67].

#### 6.3.1. Maternal Factors

In spite of the altered maternal pharmacokinetics, no studies justify a change in dosage [68]: pregnant women affected by BC should receive similar treatments to non-pregnant ones, even though modified to protect the fetus, as an adverse neonatal outcome is related to drug dose and to the timing and degree of exposure to cytotoxic drugs [69].

#### 6.3.2. Timing

Despite the limited number of studies available, the data on fetal outcomes in patients who have received chemotherapy seem reassuring. As a rule of thumb, the risk of serious effects on the fetus is low and abortion is not necessary if the diagnosis is made in the second and third trimester or if chemotherapy can be started after the 14th week of pregnancy [67].

Major recommendations during pregnancy are summarized in Figure 5. 

#### 6.3.3. Drug Factors and Fetal Effects

There are several chemotherapy regimens studied for the treatment of PABC [70]. The most used is currently Epirubicin-Cyclophosphamide, while the use of Cyclophosphamide-Methotrexate-Fluorouracile, Taxanes, and Alkaloids of vinca has been reported in a few cases. 

Unfortunately, after using Cyclophosphamide during the first trimester, malformations of the toes, eyes, low ears, and cleft palate were found. Anthracyclines are well known for their cardiotoxic effects, which depend on multiple mechanisms, including oxidative damage, changes in calcium metabolism, and activation of apoptotic pathways, which lead to a progressive deterioration of heart function [70]. The most commonly used in pregnancy are Epirubicin and Doxorubicin, while Idarubicin is contraindicated, due to higher rates of adverse fetal outcomes related to its lipophilia and a greater transplacental passage.

Limited experience is available about Taxanes in PABC. Placental P-glycoprotein transporter seems to reduce the drug passing to the fetus. According to a systematic review, 50 patients with PABC exposed to Taxanes tolerated well the drugs and showed manageable toxicities; therefore, their use can be considered in combination with cisplatin/carboplatin, when clinically indicated [71].

Improved DFS and overall survival have been reported in high-risk patients undergoing dose-dense chemotherapy regimens (especially in hormone-receptors-negative PABC), without increasing the risk of fetal and/or maternal complications in pregnant women [69].

An interesting issue that has not yet been fully investigated is the effect of pregnancy on BC outcomes in mutated BRCA patients with a known history of BC. The retrospective observational study “Safety of Pregnancy in BRCA Mutated Breast Cancer Patients” aims to improve the understanding of the prognostic impact of pregnancy after BC diagnosis in BRCA-mutated survivors. Results are expected by June 2023.

### 6.4. Hormone Therapy

Tamoxifen is often used as an adjuvant treatment in premenopausal patients with positive hormone receptors. According to current guidelines, use of non-hormonal barrier contraceptives is recommended during treatment with Tamoxifen and up to 3 months after its discontinuation. However, the effects of an inadvertent conception on the fetus are not yet fully known. The effects of Tamoxifen during human pregnancy have been described in the Lareb and INCIP databases, in the registries provided by AstraZeneca, and in a prospective analysis of pregnancies occurring during and after treatment with Trastuzumab and/or Lapatinib in patients with BC [72]. 

A total of 249 fetuses exposed to Tamoxifen during the first trimester or later have been identified in a systematic review by Buonomo and colleagues [73], with 68 live births—among which 1 case of genital tract anomalies and 1 case of Goldenhar syndrome were found.

As mentioned, due to the lack of long-term data on pediatric outcomes associated with the use of Tamoxifen during pregnancy, it is not possible to draw definitive conclusions on its use; therefore, pregnancy remains contraindicated during treatment and up to 3 months after its interruption. 

Further studies have suggested that Tamoxifen can be excreted in breast milk and, therefore, its intake during lactation has to be avoided [74].

An international multi-center study aiming to evaluate the outcome of the disease and the safety of hormone-therapy pause to allow pregnancy in women with ER-positive BC is currently ongoing and might shed light on the consequences of postponing hormone therapy. Preliminary results of this study were recently presented at the San Antonio Breast Cancer Symposium 2022: a temporary hormone-therapy interruption does not impact short-term oncologic outcomes. The study will follow-up this population until 2029 in order to confirm these results [75].

### 6.5. Target Therapy

Currently, the use of Trastuzumab in HER2-positive tumors is not recommended and should be avoided in pregnant patients [76], due to adverse events.

A meta-analysis about metastatic tumors or PABC in the adjuvant setting [77] (including 18 patients) has shown the following outcomes:Healthy babies when the drug was administered only during the first trimester;Higher prevalence of adverse birth events (57%) or stillbirths when administered during the second and third trimester;Oligohydramnios/anhydramnios as the most reported event (73.3% of patients) during the second and third trimester.

Such findings were attributed to the large molecular size of the drug, leading to poor transplacental transfer during the first trimester.

To date, safety data about the use of Pertuzumab in pregnant patients are lacking [78]. An observational cohort study, the MotHER Pregnancy Registry, is currently ongoing in the United States in women with BC who have been treated with Trastuzumab, with or without Pertuzumab, or Ado-trastuzumab emtansine during pregnancy. Results of this study will help to better estimate adverse events from these drugs during pregnancy.

### 6.6. Immunotherapy

High immunogenicity and good responsiveness to immune checkpoint inhibitors (ICIs) are features of the most common types of BC in pregnancy—namely, triple-negatives and HER2-positive BC [79]. However, as ICIs target the PD-1/PD-L1 pathway, they can interfere with maternal immune tolerance of the fetus and induce an immune response [76,80], increasing the risk of miscarriage and potentially causing serious maternal morbidity. Current drugs are IgG4 antibodies, able to cross the placenta and directly harm the fetus. In animal models, anti-PD-1/PD-L1 and anti-CTLA-4 inhibitors during pregnancy have been associated with increased rates of abortion, stillbirths, premature births, and a higher incidence of mortality, especially when used during the third trimester [81].

### 6.7. Lactation during Systemic Therapy

Breastfeeding is a pivotal moment in the mother–child relationship, and it has a great benefit on newborns. However, on the basis of current evidence, breastfeeding or lactation is not recommended while receiving systemic chemotherapy, endocrine therapy, target therapy, and immunotherapy, due to the possibility of drug excretion in breast milk [82].

## 7. Conclusions

Experiencing BC during pregnancy or lactation is a challenging and delicate situation for the mother herself, both in terms of physical and psychological aspects, and for the practitioners involved.

US is the first-line imaging technique, as it is safe for the fetus, widely available, and usually decisive for the diagnosis. Mammography is nowadays considered safe in pregnancy, and should not be avoided, when necessary. Unenhanced MRI, during pregnancy, and both unenhanced and classic contrast-enhanced imaging, during lactation, can be considered for local staging. Pregnant women with BC can be safely and efficiently treated with surgery, systemic therapy, and radiotherapy, with obvious adjustments based on cancer staging and gestational age. A multidisciplinary approach is fundamental to balance fetal well-being with maternal prognosis.

## Figures and Tables

**Figure 1 diagnostics-13-00604-f001:**
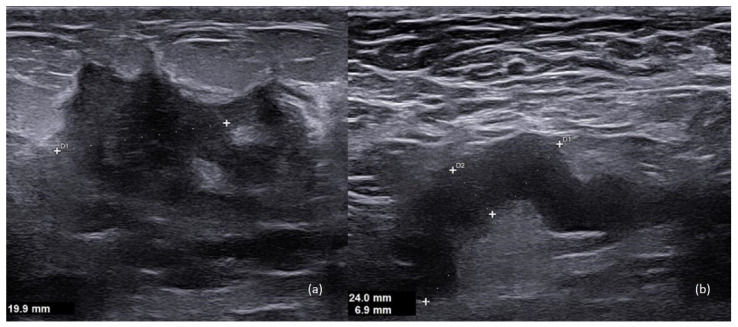
A 36-year-old patient, 6th week of pregnancy. (**a**) Hypoechoic mass with irregular margins in the inner lower quadrant of the left breast. (**b**) An ipsilateral axillary suspicious lymph node is seen, with cortical thickening (7 mm), which is an index of disease spread to the lymph nodes.

**Figure 2 diagnostics-13-00604-f002:**
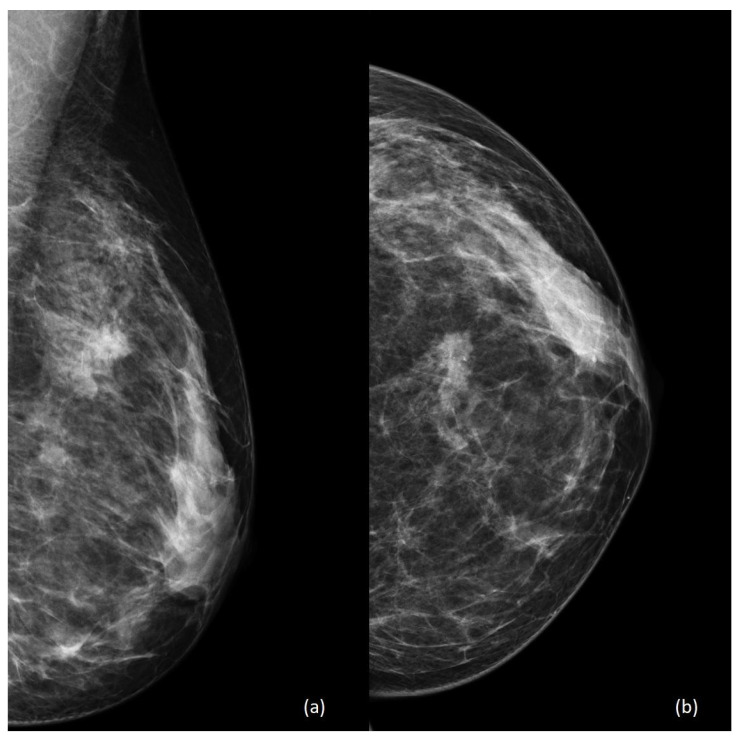
A 42-year-old patient, seventh week of pregnancy. Standard mammography including (**a**) mediolateral oblique view and (**b**) craniocaudal view of the left breast.

**Figure 3 diagnostics-13-00604-f003:**
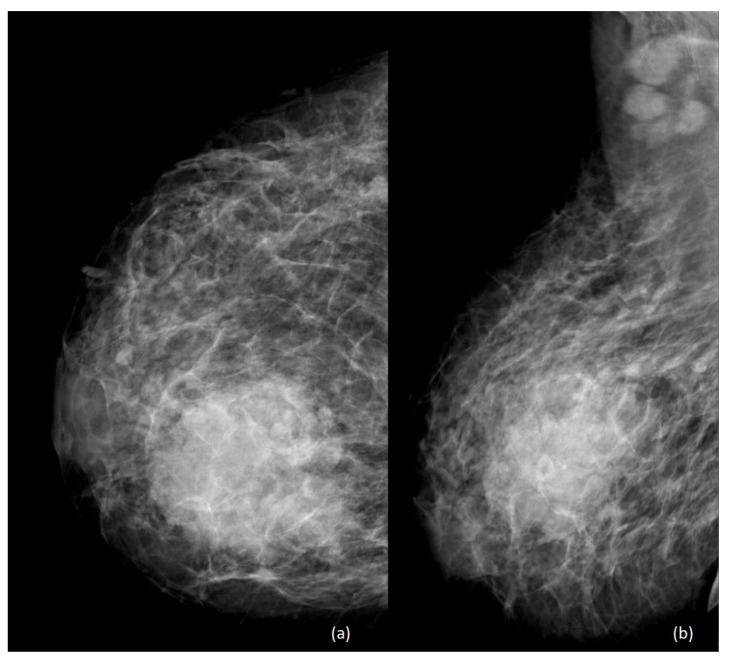
A 36-year-old breastfeeding patient, 6th month postpartum. (**a**) Craniocaudal view (**b**) and mediolateral oblique view of the right breast, characterized by increased density due to breastfeeding.

**Figure 4 diagnostics-13-00604-f004:**
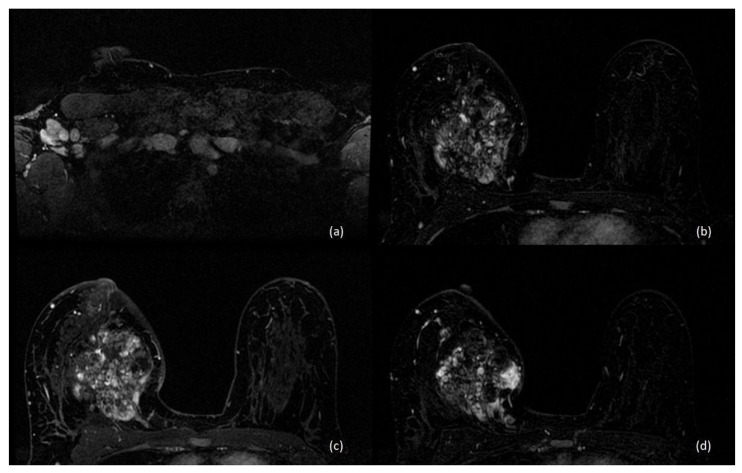
A 36-year-old breastfeeding patient, 6th month postpartum. CE-MRI of the breast shows (**a**) multiple enlarged highly suspicious lymph nodes in the right axilla and (**b**–**d**) a voluminous and heterogeneous mass enhancement, with central areas of necrosis and not circumscribed margins, located between the inner quadrants of the right breast. Histological examination revealed a G2 invasive lobular carcinoma; ER = 98%, PR = 1%, HER2-negative, Ki67 = 50% at immunohistochemistry.

**Figure 5 diagnostics-13-00604-f005:**
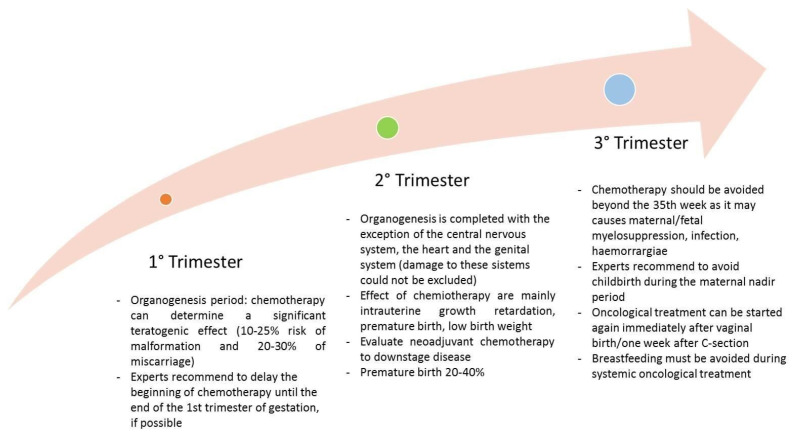
Major recommendations about systemic treatment during pregnancy and lactation.

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
