# Peer review of "Pregnancy-Associated Breast Cancer: A Diagnostic and Therapeutic Challenge"

_diagnostics, 2023, doi:10.3390/diagnostics13040604_

Round 1
Reviewer 1 Report
This paper is a very extensive survey with different approaches such as imaging modalities and treatments.
I missed some conclusions drawn in the abstract, which is very short for a so extensive manuscript. Also missed an introduction section explaining the paper's organization. The authors did some conclusions at the end of each section but there is no overall conclusion.
It is not clear to me what challenges the authors are aiming to point out and do some propositions.
Few discussions about the images. Most are in the figure caption, which is wrong from my point of view.
Regarding the section treatments, there are some subsections with few contributions such as Radiotherapy, which concludes with " As a consequence, a careful evaluation of risk-benefit ratio is essential.". This is kind of obvious for any field or treatment.
Author Response
Rev 1
This paper is a very extensive survey with different approaches such as imaging modalities and treatments.
Point 1: I missed some conclusions drawn in the abstract, which is very short for a so extensive manuscript. Also missed an introduction section explaining the paper's organization. The authors did some conclusions at the end of each section but there is no overall conclusion.
It is not clear to me what challenges the authors are aiming to point out and do some propositions.
Response 1: Dear Reviewer,
Thank you for your kind comments. Introduction and conclusion sections were added and the abstract was extended, according to your suggestion.
Point 2: Few discussions about the images. Most are in the figure caption, which is wrong from my point of view.
Response 2: Thank you for your suggestion. Captions were incorporated in the text after the description of imaging features.
Point 3: Regarding the section treatments, there are some subsections with few contributions such as Radiotherapy, which concludes with " As a consequence, a careful evaluation of risk-benefit ratio is essential". This is kind of obvious for any field or treatment.
Response 3: Thank you for your suggestion, we enriched the paragraph about radiotherapy and the other treatment subsections.
Reviewer 2 Report
In this review, the author describes pregnancy-associated breast cancer from clinical, diagnosis, and treatment aspects according to recent research status. However, there are still some problems in the manuscript. The following are comments on this work.
Major comments:
1. The author should add a conclusion chapter to summarize the full text and prospect future development.
2. The author should strengthen the transition between chapters to closely link the various parts of the paper.
3. When introducing different treatment methods, the author should emphasize the advantages and disadvantages of each method as much as possible.
Minor comments:
1. The format of this paper should be improved.
2. The authors should proofread the English writing to improve the study.
3. Author should increase the resolution of figures in the paper.
Author Response
Rev 2
In this review, the author describes pregnancy-associated breast cancer from clinical, diagnosis, and treatment aspects according to recent research status. However, there are still some problems in the manuscript. The following are comments on this work.
Major comments:
Point 1: The author should add a conclusion chapter to summarize the full text and prospect future development.
Response 1: Dear Reviewer,
Thank you for your comments and suggestions. Introduction and conclusion sections were added accordingly.
Point 2: The author should strengthen the transition between chapters to closely link the various parts of the paper.
Response 2: Thank you for your suggestion, anyway we preferred to keep the current text division in multiple chapters for the sake of clarity.
Point 3: When introducing different treatment methods, the author should emphasize the advantages and disadvantages of each method as much as possible.
Response 3: Thank you for your suggestion. We edited most of this part in order to focus on specific drawbacks of each systemic treatment and to underline that some protocols may not be employed at certain gestational ages. Specific guidelines for the pregnancy setting are not currently available in scientific literature. At present, it is mandatory for pregnant women to follow standard treatments in order to protect maternal and fetal safety. In addition, we decided to addin the hormone-therapy subsection the results of a preliminary report, which was presented at the San Antonio Breast Cancer Symposium in December 2022.
Minor comments:
- The format of this paper should be improved.
- The authors should proofread the English writing to improve the study.
- Author should increase the resolution of figures in the paper.
Thank you for your comments. The text was proofread, corrected where necessary, and modified according to your suggestions. The quality of figures was improved as well.
Reviewer 3 Report
Dear Authors,
I have reviewed your interesting manuscript entitled "Pregnancy-Associated Breast Cancer: A Diagnostic and Therapeutic Challenge"
Your manuscript is more like a report than a scientific manuscript, I strongly suggest revising it comprehensively, or discussing more the advantages and disadvantages of methods (detection and treatment)
In addition, there are newly developed techniques for the early detection of not only BC but also numerous diseases (micro RNA detection kit, qRT-PCR, biosensors used for biomarker detection, CTC cells detection, and so on), can the authors clarify why they did not have mentioned and discussed these newly methods?
BR
Author Response
Rev 3
Dear Authors,
I have reviewed your interesting manuscript entitled "Pregnancy-Associated Breast Cancer: A Diagnostic and Therapeutic Challenge"
Point 1: Your manuscript is more like a report than a scientific manuscript, I strongly suggest revising it comprehensively, or discussing more the advantages and disadvantages of methods (detection and treatment)
Response 1: Dear Reviewer,
Thank you. We rewrote most of the manuscript considering your valuable suggestion.
Point 2: In addition, there are newly developed techniques for the early detection of not only BC but also numerous diseases (micro RNA detection kit, qRT-PCR, biosensors used for biomarker detection, CTC cells detection, and so on), can the authors clarify why they did not have mentioned and discussed these newly methods?
Response 2: Thank you for this suggestion. Our first decision was to leave these newly developed techniques out of our review because they are not routinely used in everyday practice (even in non-pregnant women) and their use is limited to clinical research trails at the moment. Moreover, very few and limited research is available in literature including PABC patients. However, we decided to add a dedicated short section in the manuscript.
Round 2
Reviewer 2 Report
The authors have addressed all my comments.